# Assessing the association between air pollution and child development in São Paulo, Brazil

Ornella Luminati[1,2], Alexandra Brentani[3], Benjamin Flückiger[1,2], Bartolomeu Ledebur de Antas de Campos[1,2], Michelle Raess[1,2], Martin Röösli[1,2], Kees de Hoogh[1,2], Günther Fink[1,2]*

1 Department of Epidemiology and Public Health, Swiss Tropical and Public Health Institute, Basel, Switzerland, 2 University of Basel, Basel, Switzerland, 3 Department of Pediatrics, Medical School of São Paulo University, São Paulo, Brazil

* guenther.fink@swisstph.ch

## Abstract

### Background

Outdoor air pollution is increasingly recognised as a key threat to population health globally, with particularly high risks for urban residents. In this study, we assessed the association between residential nitrogen dioxide ($NO_2$) exposure and children's cognitive and behavioural development using data from São Paulo Brazil, one of the largest urban agglomerations in the world.

### Methods

We used data from the São Paulo Western Region Birth Cohort, a longitudinal cohort study aiming to examine determinants as well as long-term implications of early childhood development. Cross-sectional data from the 72-month follow-up was analysed. Data on $NO_2$ concentration in the study area was collected at 80 locations in 2019, and land use regression modelling was used to estimate annual $NO_2$ concentration at children's homes. Associations between predicted $NO_2$ exposure and children's cognitive development as well as children's behavioural problems were estimated using linear regression models adjusted for an extensive set of confounders. All results were expressed per 10 μg/m³ increase in NO2.

### Results

1143 children were included in the analysis. We found no association between $NO_2$ and children's cognitive development (beta -0.05, 95% CI [-0.20; 0.10]) or behavioural problems (beta 0.02, 95% CI [-0.80; 0.12]).

### Conclusion

No association between child cognition or child behaviour and $NO_2$ was found in this cross-sectional analysis. Further research will be necessary to understand the extent to which

**Data Availability Statement:** All relevant data are within the manuscript and its Supporting information files.

**Funding:** The authors received no specific funding for this work.

**Competing interests:** The authors have declared that no competing interests exist.

these null results reflect a true absence of association or other statistical, biological or adaptive factors not addressed in this paper.

## Introduction

Outdoor air pollution is increasingly recognised as one of the most critical challenges for global public health. In particular the concern arises in urban areas, where pollution levels are generally highest, since pollutants are primarily produced by the combustion of fossil fuel [1].

The most studied pollutants affecting health are particulate matter (PM), ozone ($O_3$), sulphur dioxide ($SO_2$) and nitrogen dioxide ($NO_2$). Exposure to these pollutants can trigger inflammatory processes that can cause damage to various organs such as the lungs, heart, and brain [2,3] and has also been associated with a higher risk of developing illnesses such as respiratory diseases [4], cancer [2,5] and cardiovascular problems [3,6,7].

Various studies have reported adverse health effects of air pollution for children. Prenatal and childhood exposure to $NO_2$ has been associated with asthma [8,9], failing lung development and consequent respiratory difficulties [10] and lower birth weight [10]. Existing research also suggests a positive association between traffic related air pollutants, and delays in brain maturation [11,12], memory [13,14] and cognition [12,15] among school children. Epidemiological and animal studies [16,17] have attributed these associations to changes in the central nervous system (CNS) which can be caused by an activation of the systemic [16] or local [18] immune systems and by oxidative stress [17,19].

Given that in Brazil over 80% of the population live in dense urban settings [20], average exposure to air pollution among Brazilian citizens is high [21]. São Paulo constitutes the largest urban area in Brazil and ranks among the largest cities in the world, with an estimated metropolitan population of 22 million and continued rapid growth [22]. Despite municipal efforts to improve air quality [23], air pollution levels remain a concern.

Previous studies from São Paulo have linked air pollution to non-accidental and cardiovascular mortality [7], especially in the elderly population [24], hospitalisations for respiratory problems in children [25], low birth weight [26] and premature births [27].

Few studies have investigated the impact of air pollution on cognitive development in children, most of which focus on Europe. Given that children are likely to be more susceptible to air pollution because their central nervous and immune systems are still developing [2,3,16], the lack of evidence in this area is somewhat surprising.

This study aims to address this research gap by exploring the association between nitrogen dioxide exposure and children's cognitive and behavioural development using prospectively collected data on child development and air pollution from São Paulo, Brazil.

## Methods

### Study population and study area

Data on child development was collected through the São Paulo Western Region Birth Cohort (SP-ROC) [28]. The SP-ROC cohort was set up as a longitudinal study to assess the impact of social and environmental factors on child development. All children born at the University Hospital of São Paulo between April 1, 2012 and March 31, 2014 and living in the Butantã-Jaguaré area were included in the original sample. Data was collected at birth from medical records and then during five visits: postpartum, at 6, 12, 36 and 72 months. The Butantã-Jaguaré district is about 70km$^2$ large and situated in the Western part of São Paulo city, with

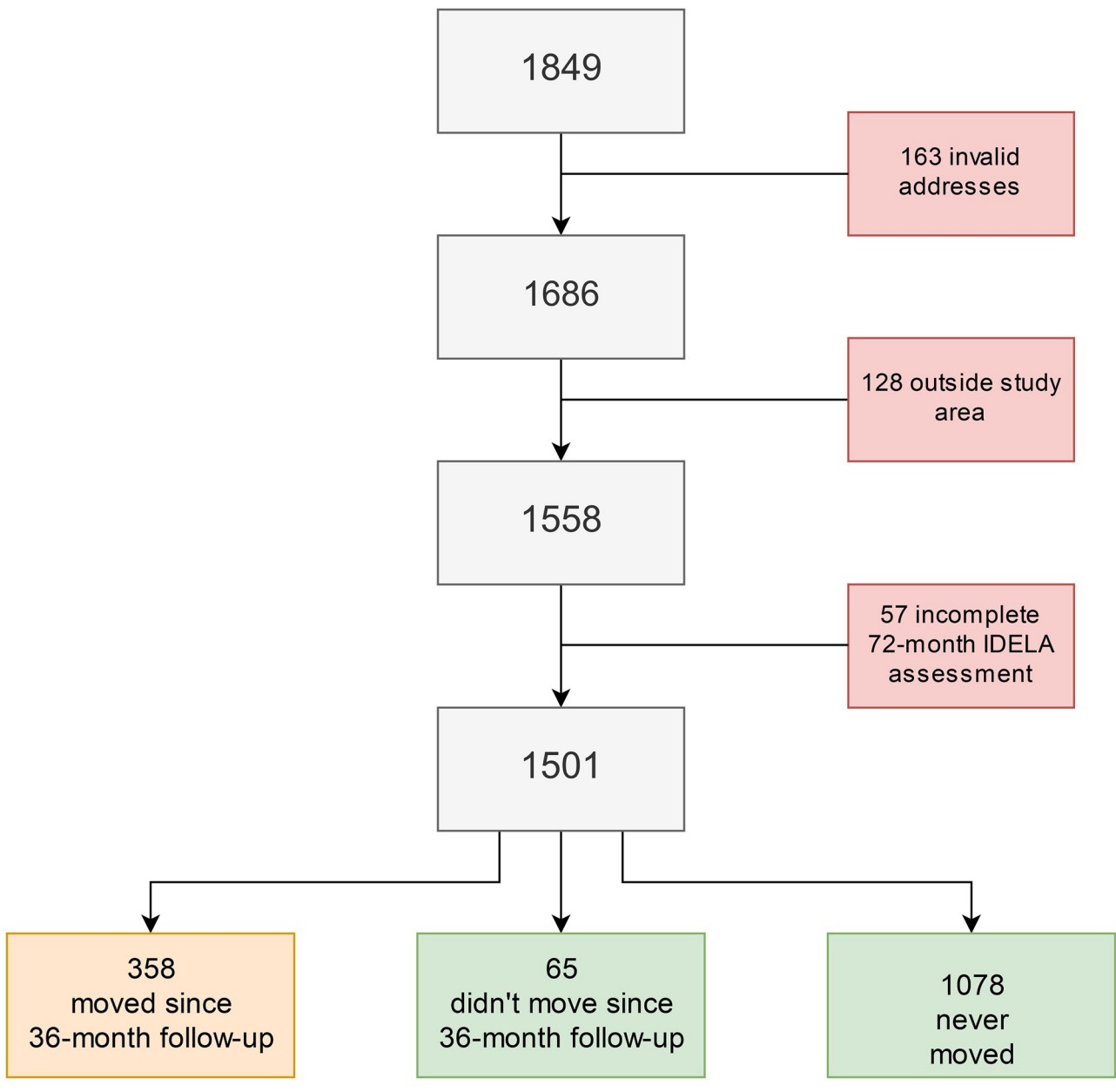

**Fig 1. Flow chart displaying the sample size.**

an estimated population of 637'000 people in 2010 [29]. The study area is characterized by a dense network of streets of all sizes and includes a wide variety of different neighbourhoods, ranging from the university campus, residential neighbourhoods and green areas to favelas and industrial areas [29].

In this study, all cohort children still living in the Butantã-Jaguaré area with a complete 72-month follow-up and geocoded address were included, as shown in the flowchart in Fig 1. For the 72-month assessment, trained interviewers surveyed 1849 children. Due to the Covid-19 situation in Brazil, data collection for this follow-up had to be terminated in April 2020. A total of 163 records had invalid addresses for geocoding and were dropped from the analysis. Additionally, 128 children were living outside the study area, while for 57, the International

Development and Early Learning Assessment (IDELA) was incomplete. Furthermore, 358 children moved between the 36-month assessment and the 72-month assessment and were excluded from the analysis because assigning appropriate $NO_2$ exposure levels was not possible. This resulted in 1143 children being eligible for the main analysis. 65 children out of these 1143 moved before the 36-month assessment and 1078 never moved.

The data collection for the SP-ROC cohort was approved by the University of São Paulo's Hospital das Clínicas ethics committee (CAPPESQ HC_FMUSP) under protocol N˚ 01604312.1.0000.0065. Written informed consent from the caregivers was obtained before data collection. The work presented in this paper was formally reviewed and approved by the Ethikkommission Nordwest- und Zentralschweiz (EKNZ) under protocol N˚ AO_2020–00024.

## $NO_2$ exposure

With the goal to investigate the effect of air pollution on child development, we focused on $NO_2$ as our primary pollutant. $NO_2$ primarily results from traffic related burning of fossil fuel and can be measured relatively easily and cheaply using passive gas samplers (80). Annual mean outdoor $NO_2$ concentrations at the home address of the children were predicted by a land use regression (LUR) model based on $NO_2$ measurements over 2 one-week periods at 80 locations in 2019. The first measurements were done in summer (February), reflecting the hot humid and rainy conditions, the second in winter (August) reflecting the colder and dry conditions. Temporal adjustments were made using weekly data collected at a reference monitoring site all year around to calculate adjusted annual mean $NO_2$ concentrations. As described in more details elsewhere [30], a LUR model was developed, using supervised stepwise linear regression by including GIS predictor variables (e.g. roads, land use, altitude, green space etc.), explaining 66% of the variation in the adjusted annual mean $NO_2$ concentrations. The predictions from the annual model were then mapped for the entire study area on a 25x25 meters grid in QGIS 3.4.4. Using the same software, the predicted $NO_2$ values were assigned to each child's geocoded home address.

## Outcome measures

Based on the existing literature, we considered two primary outcomes: children's cognitive ability and children's behavioural difficulties.

**Cognition.** Children's overall development was assessed using the *International Development and Early Learning Assessment (IDELA)* tool. The IDELA includes 22 tasks divided into the following four domains: Motor (gross and fine), Communication/Emerging Literacy, Problem solving/Emergent Numeracy, and Personal-social/Social-emotional [31]. Trained interviewers explained the tasks to each child and recorded observed answers. Raw IDELA scores were standardised to z-scores within the analysed sample.

**Behavioural problems.** Children's behavioural problems were captured using the *Child Behavior Checklist (CBCL)* [32]. The CBCL scale contains 120 items for children aged 6 to 18 and are graded on a three-point Likert scale, where 0 means that the behaviour was not reported by the caregiver (absent), 1 means the behaviour occurs sometimes, and 2 means the behaviour occurs often. All questions ask about children's behaviour within the six months preceding the interview. In order to facilitate interpretation, we normalised raw CBCL scores within our sample to a z-score with mean zero and standard deviation of one.

Caregiver interview and child assessments were performed during a home visit by a trained team of enumerators with previous experience in child assessments. Before assessment, caregivers signed a written consent form.

## Statistical analysis

In order to describe the sample regarding demographic, socioeconomics and other variables, we firstly performed descriptive statistics. The main analysis, was a cross-sectional analyses of the outcome data collected during the 72-month follow-up. We standardised both outcomes to mean zero and standard deviation one. We executed univariate and multivariable linear regressions to check for confounders. The univariate and the multivariate models are given by:

**Univariate Model**

$$Y_i = \beta_0 + \beta_1 \ NO_2 + \varepsilon_i$$

**Multivariate Model**

$$\begin{aligned} Y_i = \beta_0 &+ \beta_1 \ NO_2 + \beta_2 \ \mathrm{gender} + \beta_3 \ \mathrm{age\_months} + \beta_4 \ \mathrm{skincolor} + \beta_5 \ \mathrm{birthweight} \\ &+ \beta_6 \ \mathrm{gest\_length} + \beta_7 \ \mathrm{delivery} + \beta_8 \ \mathrm{age\_mother} + \beta_9 \ \mathrm{mother\_skincolor} \\ &+ \beta_{10} \ \mathrm{mother\_depression} + \beta_{11} \ \mathrm{caregiver\_age} + \beta_{12} \ \mathrm{marital\_status} \\ &+ \beta_{13} \ \mathrm{caregiver\_relation} + \beta_{14} \ \mathrm{caregiver\_grade} + \beta_{15} \ \mathrm{headhousehold\_grade} \\ &+ \beta_{16} \ \mathrm{household\_size} + \beta_{17} \ \mathrm{financial\_support} + \beta_{18} \ \mathrm{SES} + \beta_{19} \ \mathrm{MICS} + \varepsilon_i \end{aligned}$$

Where $Y_i$ is the outcome of an child i, $NO_2$ is the estimated annual exposure, and $\varepsilon_i$ is the model residuals. To address heteroscedasticity as well as potential non-linearity concerns, we used the robust sandwich estimator. For the sensitivity analysis linear regressions were performed on a sub sample. Only children living at the same address since birth were considered in the subsample. Additionally, we performed linear regressions with a categorical exposure variable to compare children exposed to $NO_2$ concentrations in the bottom and the top decile with the rest of the sample. As a further sensitivity analysis, we repeated the main analysis by omitting the children considered being outliers. Children showing an IDELA z-score lower than -3.5 (11 children) and children showing a CBCL z-score higher than 3.75 (6 children) were considered outliers. All statistical analyses were performed using the STATA version 16.0 statistical software package [33].

## Other variables

We adjusted for an extensive list of potential confounders in the regression, including child gender, child age in months, child skin-colour, birthweight, gestational length, delivery type, mother's age at delivery, mother's skin-colour, maternal depression, caregiver's marital status, caregiver's relation to the child, caregiver's age, highest school grades of caregiver, highest school grade of household head, household size, financial support, socio-economic status (following the Brazilian economic status classifications) and home stimulation score.

Apart from gender, birthweight, gestational length and delivery type, which were collected at birth; all data were collected by the SP-ROC-Cohort 72-month follow-up questionnaire.

## Results

A total of 1143 children were included in our main analysis. As shown in Table 1, 51% of children were male. The age at assessment varied between 4 to 7.5 years, with a mean age of 76 months. Most of the children were classified as white (45%) or mixed (51%). 81 children were born preterm (7%) and 6% had a birth weight < 2500 grams. Mother's age at delivery ranged between 13 and 46 years, with a median of 26 years. 37% of children were born with a

**Table 1. Participants' characteristics.**

| Variables | | N (Percentage) | |
|---|---|---|---|
| **Female gender** | | 558 | (48.82) |
| **Age in months**[*] | | 76 | ± 5.20 |
| **Child's skin-color** | White | 517 | (45.23) |
| | Mixed | 581 | (50.83) |
| | Black | 42 | (3.67) |
| | Others | 3 | (0.26) |
| **Low weight at birth (<2500g)** | | 74 | (6.47) |
| **Pre-term gestation** | | 81 | (7.10) |
| **Delivery type** | Regular | 545 | (47.68) |
| | Caesarean | 423 | (37.01) |
| | Forceps | 175 | (15.31) |
| **Mother's age at delivery** | ≤19 | 172 | (15.05) |
| | 20–29 | 609 | (53.28) |
| | ≥30 | 362 | (31.67) |
| **Mother having depression** | | 500 | (43.74) |
| **Caregiver is married or live with a partner** | | 729 | (64.63) |
| **Caregiver's highest grade completed** | None | 23 | (2.02) |
| | Elementary | 342 | (30.08) |
| | Middle | 656 | (57.70) |
| | Upper | 116 | (10.20) |
| **Households getting financial support** | | 272 | (23.80) |

Based on 1143 participants. Mean and standard deviation (rather than N%)shown for age.

caesarean. At the 72-month follow up 65% of the caregivers lived with a partner and 68% completed a middle or higher school grade. 24% of the households received financial support. 44% of the mothers showed symptoms of mild or severe depression. Further characteristics are described in S1 Table. The comparison of the sample at birth and the sample at the 72-month follow-up is presented in S2 Table. Although we had a high rate of loss of follow-up, mostly because of the Covid-19 pandemic, the sample's characteristics did not differ from the original sample at birth.

Mean annual $NO_2$ concentration ranged from 31.1 μg/m$^3$ to 128.8 μg/m$^3$ in the study area. Estimated concentrations at the residential addresses of children ranged from 34.0 μg/m$^3$ to 98.0 μg/m$^3$, with a median exposure of 40.8 μg/m$^3$ and a standard deviation of 4.9 μg/m$^3$. All participants were exposed to $NO_2$ concentrations higher than 10 μg/m$^3$, the WHO air quality guideline value for annual mean $NO_2$ [34].

IDELA and CBCL z-scores ranged between -5.98 and 1.40 (IDELA) and between -1.15 and 5.69 (CBCL).

Table 2 provides the main regressions' results for the full sample of 1143 children, as well as for the subsample of children who never moved (N = 1078). We found no significant associations between $NO_2$ and children's cognition (IDELA) and behaviour (CBCL) in either sample.

Table 3 shows results for a categorical exposure variable. No indications for decreased cognitive functions or increased behavioural problems for the highest exposed group (highest decile) was found. Fig 2 provides further details on the exposure levels in the higher and lower decile.

**Table 2. Association of NO$_2$ exposure as continuous [μg/m$^3$] with IDELA (z-score) and with CBCL (z-score), unadjusted and adjusted* models.**

| | Unadjusted | | N | Adjusted | | N |
|---|---|---|---|---|---|---|
| **IDELA z-score** | **β (95% CI)** | **p-value** | | **β (95% CI)** | **p-value** | |
| **All**** | -0.05 (-0.19;0.10) | 0.52 | 1143 | -0.05 (-0.20;0.10) | 0.52 | 1098 |
| **Never Moved**\*\*\* | -0.08 (-0.23;0.07) | 0.30 | 1078 | -0.08 (-0.24;0.07) | 0.28 | 1033 |
| **CBCL z-score** | | | | | | |
| **All** | 0.003 (-0.09;0.10) | 0.95 | 1142 | 0.02 (-0.80;0.12) | 0.75 | 1098 |
| **Never Moved** | 0.02 (-0.09;0.12) | 0.75 | 1077 | 0.03 (-0.08;0.14) | 0.61 | 1033 |

Results are expressed per 10 μg/m3.

* Models adjusted for: child gender, child age in months, child skin-color, birthweight, gestational length, delivery type, mother's age at delivery, mother's skin-color, maternal depression, caregiver's marital status, caregiver's relation to the child, caregiver's age, highest school grades of caregiver, highest school grade of household head, household size, financial support, socio-economic status, and home stimulation score.

**All: These children live at the same address since longer than three years before 72-moth follow-up.

\*\*\*Never moved: These children never changed the address since birth. These children constitute the sensitivity analysis sample.

IDELA = International Development and Early Learning Assessment; CBCL = Child Behavior Checklist; CI = Confidence interval.

In S3 Table we show results that exclude outliers—the results are consistent with the full sample regressions presented here.

## Discussion and conclusion

This study aimed to assess association between exposure to NO$_2$ and child development. Overall exposure levels were very high in the study settings, with all participants exposed to NO$_2$ concentrations above the WHO annual recommended level of 10 μg/m$^3$ [34]. Despite these high levels of exposure, we found no association between exposure to NO$_2$, and children's cognitive development or children's behavioural problems in our sample.

In the literature, evidence on the association between NO$_2$ and child development remains uncertain. Freire et al. found a negative association between NO$_2$ exposure higher than 24 μg/m$^3$ and general cognition compared with NO$_2$ exposures lower than 15.4 μg/m$^3$ [35], although these differences were not statistically significant. Forns et al. looked at air pollution in

**Table 3. Unadjusted and adjusted* association of bottom and top decile exposure to NO$_2$ [μg/m3] with IDELA (z-score) and CBCL (z-score).**

| | Unadjusted | | N | Adjusted | | N |
|---|---|---|---|---|---|---|
| | **β (95% CI)** | **p-value** | | **β (95% CI)** | **p-value** | |
| **IDELA z-score** | | | 1143 | | | 1098 |
| **38.9–45.7** | REF | | 915 | REF | | 879 |
| **<38.9** | -0.01 (-0.20;0.19) | 0.93 | 114 | -0.09 (-0.28;0.10) | 0.37 | 109 |
| **>45.7** | 0.03 (-0.19;0.26) | 0.76 | 114 | -0.01 (-0.24;0.23) | 0.97 | 110 |
| **CBCL z-score** | | | 1142 | | | |
| **38.9–45.7** | REF | | 914 | REF | | 1098 |
| **<38.9** | 0.14 (-0.06;0.35) | 0.17 | 114 | 0.11 (-0.08;0.31) | 0.26 | 109 |
| **>45.7** | 0.03 (-0.13;0.19) | 0.70 | 114 | 0.06 (-0.10;0.22) | 0.44 | 110 |

*Models adjusted for: child gender, child age in months, child skin-color, birthweight, gestational length, delivery type, mother's age at delivery, mother's skin-color, maternal depression, caregiver's marital status, caregiver's relation to the child, caregiver's age, highest school grades of caregiver, highest school grade of household head, household size, financial support, socio economic status, and home stimulation score.

IDELA = International Development and Early Learning Assessment; CBCL = Child Behavior Checklist; CI = Confidence interval.

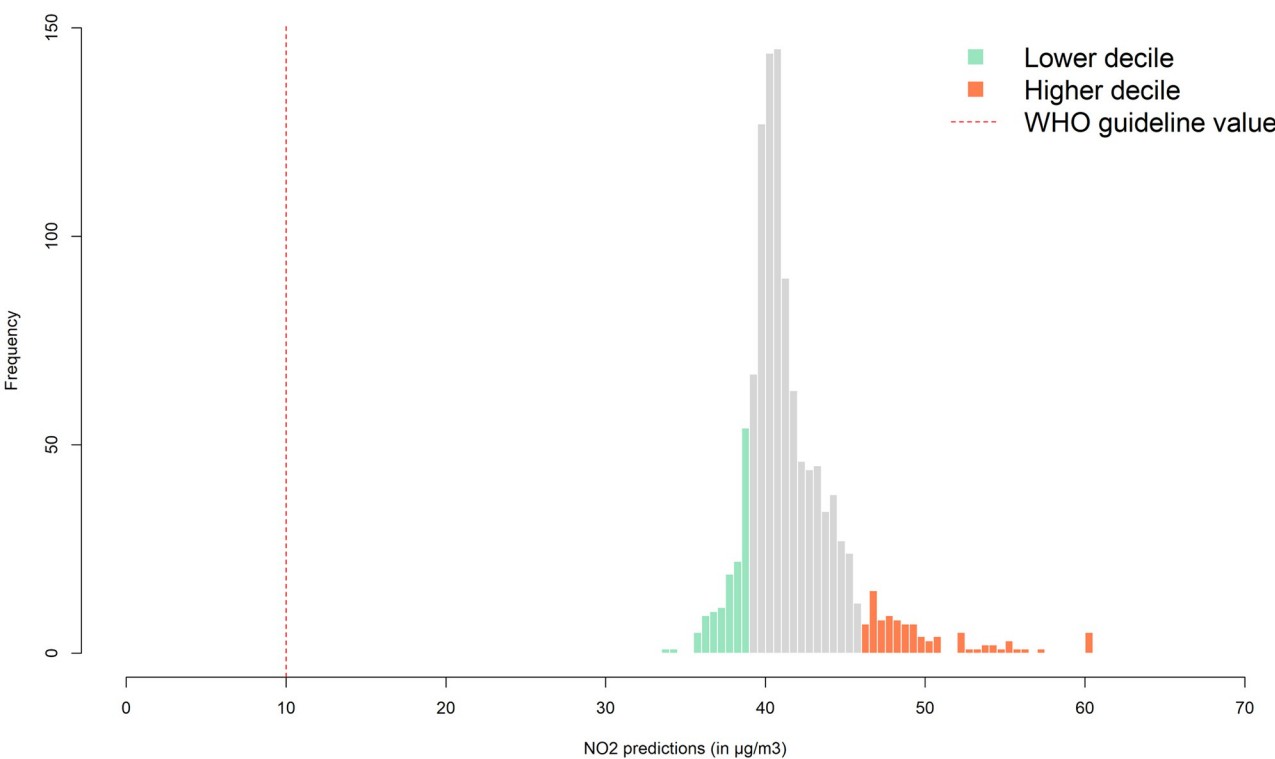

**Fig 2. Range and frequency of predicted NO$_2$ concentrations at children's home addresses.**

different schools in Barcelona and found a significant negative association with all measured pollutants and working memory, and largest associations for NO$_2$ [13]. Comparing children at high and low NO$_2$ exposure Sunyer et al. found negative associations between NO$_2$ and working memory [14]. In general, most of the literature seems to find a negative relationship between NO$_2$ exposure and children's cognition, although the overall evidence was classified as insufficient in literature review from 2015 [36]. The same literature review also reported insufficient evidence for the association between NO$_2$ and child behaviour [36]. The majority of studies published so far have focused on exposure levels at school [13,14].

This project was initiated with a strong prior of finding negative associations between NO$_2$ exposure and children's development and behaviour. We performed NO2 measurements directly in the study area, within the personal premises of 80 SP-ROC study participants. These measurements enabled us to build a robust LUR model to predict NO2 concentrations. Another strength of our study is the SP-ROC-Cohort dataset, which contains detailed information on families and home environments, enabling us control for a large range of potential confounders. The data set also contains both direct assessments of children's cognitive skills and detailed reports on behavioural issues, allowing us to look at both dimensions of child development.

Despite this setup, we found no evidence of negative associations between NO$_2$ and child's development and behaviour. There are some limitations that may have contributed to the lack of association. First, we estimated the NO$_2$ concentration at the residential address of the children, which describes their exposure only during a part of the day. Most of the children in the study went to kindergarten or school, and we did not collected data on NO$_2$ concentration in these locations. Second, as with all NO$_2$ exposure models, a certain extent of misclassification

of exposure levels cannot be avoided. Given that we measured exposure levels twice across 80 different measurement locations, the scope for such measurement error seems however small. From a purely statistical perspective, we were of course also limited by the very high average exposure levels with relatively limited variability. The sample contained only 2 children with $NO_2$ exposure levels below 35 μg/m$^3$, and only a few with $NO_2$ exposure levels above 50 μg/m$^3$. Even if $NO_2$ exposure is harmful, it is possible that differences in exposure levels in our sample were too small to result in statistically detectable differences in child outcomes. It is of course also possible that the true causal effect of $NO_2$ exposure is smaller than the current literature suggests, or that negative effects only occur in settings where other protective factors (that may have been in place here) are not present.

Further research will be needed to better understand the diverging results in the current literature.

## Supporting information

**S1 Table. Participants' characteristics at 72-month follow-up.**
(DOCX)

**S2 Table. Participants' characteristics at birth vs. at 72-month follow-up.**
(DOCX)

**S3 Table. Sensitivity analysis without outliers.**
(DOCX)

**S1 Dataset.**
(CSV)

**S1 File.**
(DOCX)

## Acknowledgments

We would like to thank the Eckenstein-Geigy Professorship for supporting the collection of the $NO_2$ data.

## Author Contributions

**Conceptualization:** Ornella Luminati, Martin Röösli, Kees de Hoogh, Günther Fink.

**Data curation:** Ornella Luminati, Benjamin Flückiger, Bartolomeu Ledebur de Antas de Campos, Michelle Raess, Günther Fink.

**Formal analysis:** Ornella Luminati, Kees de Hoogh, Günther Fink.

**Funding acquisition:** Günther Fink.

**Investigation:** Alexandra Brentani.

**Methodology:** Ornella Luminati, Martin Röösli, Kees de Hoogh, Günther Fink.

**Resources:** Alexandra Brentani.

**Software:** Benjamin Flückiger, Kees de Hoogh.

**Supervision:** Martin Röösli, Kees de Hoogh, Günther Fink.

**Visualization:** Ornella Luminati.

**Writing – original draft:** Ornella Luminati.

**Writing – review & editing:** Alexandra Brentani, Benjamin Flückiger, Bartolomeu Ledebur de Antas de Campos, Michelle Raess, Martin Röösli, Kees de Hoogh, Günther Fink.

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
