## [Decision Letter · Decision Letter 0]

10 Nov 2021

PONE-D-21-30740Assessing the association of air pollution and child development in São Paulo, BrazilPLOS ONE

Dear Dr. Fink,

Thank you for submitting your manuscript to PLOS ONE. After careful consideration, we feel that it has merit but does not fully meet PLOS ONE’s publication criteria as it currently stands. Therefore, we invite you to submit a revised version of the manuscript that addresses the points raised during the review process.

We look forward to receiving your revised manuscript.

Kind regards,

Flavio Manoel Rodrigues Da Silva Júnior

Academic Editor

PLOS ONE

4. We note that Figure 2 in your submission contain map images which may be copyrighted. All PLOS content is published under the Creative Commons Attribution License (CC BY 4.0), which means that the manuscript, images, and Supporting Information files will be freely available online, and any third party is permitted to access, download, copy, distribute, and use these materials in any way, even commercially, with proper attribution. For these reasons, we cannot publish previously copyrighted maps or satellite images created using proprietary data, such as Google software (Google Maps, Street View, and Earth). For more information, see our copyright guidelines: http://journals.plos.org/plosone/s/licenses-and-copyright.

Reviewers' comments:

Reviewer's Responses to Questions

**Comments to the Author**

1. Is the manuscript technically sound, and do the data support the conclusions?

Reviewer #1: Partly

Reviewer #2: Partly

2. Has the statistical analysis been performed appropriately and rigorously? 

Reviewer #1: No

Reviewer #2: Yes

3. Have the authors made all data underlying the findings in their manuscript fully available?

Reviewer #1: Yes

Reviewer #2: No

4. Is the manuscript presented in an intelligible fashion and written in standard English?

Reviewer #1: Yes

Reviewer #2: Yes

5. Review Comments to the Author

Reviewer #1: This work is a useful addition to the literature. However, there are some instances where the author needs to make some improvements. Below are my comments:

1) The authors should highlight the methodology's appropriateness. Why linear regression model was used? I have doubts that this is the most appropriate statistical approach. Moreover, in Methods section, it’s important to describe how the model and metrics were applied, including their equations.

2) The influence of meteorological conditions on the air pollutant concentrations needs to be considered at the time of the study.

3) Discussion: comparisons with previous studies are absent.

Reviewer #2: The present study addressed the association between nitrogen dioxide exposure and children’s cognitive and behavioral development in São Paulo City. It is a relevant theme and the importance of the study is clearly stated in the manuscript. However, a substantial modification is required and some issues must be addressed:

I suggest to include more information regarding the NO2 when the authors introduced the air pollutants, considering it is the pollutant included in the study.

Why did the authors choose to monitor the NO2 concentration and why the concentration of other pollutants was not included?

I suggest a reorganization of the introduction section. Some paragraphs could be merged to make the text more concise.

The characterization of the study area could be more specific, including the main pollution sources nearby and a figure to illustrate.

Why did the authors choose a 2 one-week monitoring period in the summer and winter, This information could be better explained in the methods section.

Line 225 – the sentence “ This study aimed at assessing the associations between exposure to NO2 and children’s 226 development.” should be grammar checked.

To continue the revision process of this article, the authors should state clearly the contribution to the results to the literature, make a more robust discussion about their data, and demonstrate how the strengths of the study minimize the limitations observed.

The author needs to rearrange the conclusions, once the information in the conclusion is still not very informative and do not reflect the content and results of the research that has been done.

The quality of the figures should be improved and the format of the tables revised.

The manuscript must be grammar-checked.

6. PLOS authors have the option to publish the peer review history of their article (what does this mean?). If published, this will include your full peer review and any attached files.

Reviewer #1: No

Reviewer #2: No

---

## [Author Response · Author response to Decision Letter 0]

18 Jan 2022

Reviewer #1: This work is a useful addition to the literature. However, there are some instances where the author needs to make some improvements. Below are my comments:

1) The authors should highlight the methodology's appropriateness. Why linear regression model was used? I have doubts that this is the most appropriate statistical approach. Moreover, in Methods section, it’s important to describe how the model and metrics were applied, including their equations.

Thank you for this comment and apologies for the lack of clarity in the original manuscript. The linear model was used because we are working with a normalized outcome variable, i.e. z-score with mean 0 and SD 1. In the text we added following sentence to clarify the appropriateness of the method: “We standardised both outcomes to mean zero and standard deviation one.” Line 146-147. We also added the following equations to the text, which describe the exact model used in the regressions.

Y_i= β_0 + β_(1 ) 〖NO〗_2 + ε_i 

Y_i= β_0 + β_(1 ) 〖NO〗_2 + β_(2 ) gender + β_(3 ) age_months+ β_(4 ) skincolor+ β_(5 ) birthweight+ β_(6 ) gest_length+ β_(7 ) delivery+ β_(8 ) age_mother+ β_(9 ) mother_skincolor+ β_(10 ) mother_depression+ β_(11 ) caregiver_age+ β_(12 ) marital_status+ β_(13 ) caregiver_relation+ β_(14 ) caregiver_grade+ β_(15 ) headhousehold_grade+ β_(16 ) household_size+ β_(17 ) financial_support + β_(18 ) SES + β_(19 ) MICS+ ε_i

2) The influence of meteorological conditions on the air pollutant concentrations needs to be considered at the time of the study.

Many thanks for this comment. We took differences in meteorological conditions in our air pollution modelling into account by performing the NO2 monitoring campaigns in 2 different seasons; winter (August 2019) reflecting hot, humid and rainy conditions and summer (Feb 2019) reflecting the dry conditions. At the same time we measured NO2 continuously (weekly NO2 measurements) at a regional background site inside the study area over a full year. Using this continuous record, we were able to adjust the summer and winter NO2 measurements from the 2 campaigns to an adjusted annual average NO2 concentration. The details of the monitoring campaign and the subsequent calculation of the adjusted annual average NO2 concentrations are described in Luminati et al. (2021). In the revised text we integrated this comment on lines 112-115: 

“The first measurements were done in summer (February), reflecting the hot humid and rainy conditions, the second in winter (August) reflecting the colder and dry conditions. Temporal adjustments were made using weekly data collected at a reference monitoring site all year around to calculate adjusted annual mean NO2 concentrations.”

3) Discussion: comparisons with previous studies are absent.

We carefully edited the discussion. 

Reviewer #2: The present study addressed the association between nitrogen dioxide exposure and children’s cognitive and behavioral development in São Paulo City. It is a relevant theme and the importance of the study is clearly stated in the manuscript. However, a substantial modification is required and some issues must be addressed:

Many thanks for these friendly comments. We did our best to address all remaining issues as outlined below.

I suggest to include more information regarding the NO2 when the authors introduced the air pollutants, considering it is the pollutant included in the study. Why did the authors choose to monitor the NO2 concentration and why the concentration of other pollutants was not included? 

We focused on NO2 as this is a good proxy for traffic related air pollution and it is relatively easy and cheap to monitor. We have added this information to the text on lines 107-109 where we write:

“With the goal to investigate the effect of air pollution on child development, we focused on NO2 as our primary pollutant. NO2 primarily results from traffic related burning of fossil fuel and can be measured relatively easily and cheaply using passive gas samplers (80).“

I suggest a reorganization of the introduction section. Some paragraphs could be merged to make the text more concise.

We did some changes in the introduction in order to make the text more concise. 

The characterization of the study area could be more specific, including the main pollution sources nearby and a figure to illustrate. 

We added some more information in the text addressing this comment as suggested. On lines 84-87 of the revised manuscript, we write:

“The study area is characterized by a dense network of streets of all sizes and includes a wide variety of different neighbourhoods, ranging from the university campus, residential neighbourhoods and green areas to favelas and industrial areas (29).” 

Why did the authors choose a 2 one-week monitoring period in the summer and winter, This information could be better explained in the methods section.

Thank you. We integrated more information in the revised text on lines 112-115, where we write: ”The first measurements were done in summer (February), reflecting the hot humid and rainy conditions, the second in winter (August) reflecting the colder and dry conditions.”

To continue the revision process of this article, the authors should state clearly the contribution to the results to the literature, make a more robust discussion about their data, and demonstrate how the strengths of the study minimize the limitations observed. The author needs to rearrange the conclusions, once the information in the conclusion is still not very informative and do not reflect the content and results of the research that has been done.

We carefully edited the discussio

---

## [Decision Letter · Decision Letter 1]

10 Mar 2022

PONE-D-21-30740R1Assessing the association between air pollution and child development in São Paulo, BrazilPLOS ONE

Dear Dr. Fink,

Thank you for submitting your manuscript to PLOS ONE. After careful consideration, we feel that it has merit but does not fully meet PLOS ONE’s publication criteria as it currently stands. Therefore, we invite you to submit a revised version of the manuscript that addresses the points raised during the review process.

We look forward to receiving your revised manuscript.

Kind regards,

Flavio Manoel Rodrigues Da Silva Júnior

Academic Editor

PLOS ONE

Reviewers' comments:

Reviewer's Responses to Questions

**Comments to the Author**

1. If the authors have adequately addressed your comments raised in a previous round of review and you feel that this manuscript is now acceptable for publication, you may indicate that here to bypass the “Comments to the Author” section, enter your conflict of interest statement in the “Confidential to Editor” section, and submit your "Accept" recommendation.

Reviewer #2: (No Response)

2. Is the manuscript technically sound, and do the data support the conclusions?

Reviewer #2: Partly

3. Has the statistical analysis been performed appropriately and rigorously? 

Reviewer #2: No

4. Have the authors made all data underlying the findings in their manuscript fully available?

Reviewer #2: Yes

5. Is the manuscript presented in an intelligible fashion and written in standard English?

Reviewer #2: Yes

6. Review Comments to the Author

Reviewer #2: The authors made the modifications in the introduction, however, the discussion is still insufficient to make the result more robust and it is not well explored. The authors had an interest and complete databases, however, they could not discuss the absence of result and attributed only to limitations, as the lack of data and sample.

7. PLOS authors have the option to publish the peer review history of their article (what does this mean?). If published, this will include your full peer review and any attached files.

Reviewer #2: No

---

## [Author Response · Author response to Decision Letter 1]

17 Mar 2022

Reviewer #2 Comments:

“The authors made the modifications in the introduction, however, the discussion is still insufficient to make the result more robust and it is not well explored. The authors had an interest and complete databases, however, they could not discuss the absence of result and attributed only to limitations, as the lack of data and sample.”

(Reply): Thanks for the kind review. We are happy to hear that the introduction is good shape now. We have done our best to further improve the Discussion. We agree that we have a rather nice and comprehensive database, but that unfortunately does not allow us to directly identify the reason why we do not see the expected outcome gradient. There are many potential reasons why this could be the case, and we can only speculate about them. We pasted the revised Discussion below and would be grateful for any further suggestions.

“Despite these high levels of exposure, we found no association between exposure to NO2, and children’s cognitive development or children’s behavioural problems in our sample. In the literature, evidence on the association between NO2 and child development remains uncertain. Freire et al. found a negative association between NO2 exposure higher than 24 μg/m3 and general cognition compared with NO2 exposures lower than 15.4 μg/m3 (37), although these differences were not statistically significant. Forns et al. looked at air pollution in different schools in Barcelona and found a significant negative association with all measured pollutants and working memory, and largest associations for NO2 (13). Comparing children at high and low NO2 exposure Sunyer et al. found negative associations between NO2 and working memory (14). In general, most of the literature seems to find a negative relationship between NO2 exposure and children’s cognition, although the overall evidence was classified as insufficient in literature review from 2015 (38). The same literature review also reported insufficient evidence for the association between NO2 and child behaviour (38). The majority of studies published so far have focused on exposure levels at school (13, 14, 39).

This project was initiated with a strong prior of finding negative associations between NO2 exposure and children’s development and behaviour. We performed NO2 measurements directly in the study area, within the personal premises of 80 SP-ROC study participants. These measurements enabled us to build a robust LUR model to predict NO2 concentrations. Another strength of our study is the SP-ROC-Cohort dataset, which contains detailed information on families and home environments, enabling us control for a large range of potential confounders. The data set also contains both direct assessments of children’s cognitive skills and detailed reports on behavioural issues, allowing us to look at both dimensions of child development.

Despite this setup, we found no evidence of negative associations between NO2 and child’s development and behaviour. There are some limitations that may have contributed to the lack of association. First, we estimated the NO2 concentration at the residential address of the children, which describes their exposure only during a part of the day. Most of the children in the study went to kindergarten or school, and we did not collected data on NO2 concentration in these locations. Second, as with all NO2 exposure models, a certain extent of misclassification of exposure levels cannot be avoided. Given that we measured exposure levels twice across 80 different measurement locations, the scope for such measurement error seems however small. From a purely statistical perspective, we were of course also limited by the very high average exposure levels with relatively limited variability. The sample contained only 2 children with NO2 exposure levels below 35 μg/m3, and only a few with NO2 exposure levels above 50 μg/m3. Even if NO2 exposure is harmful, it is possible that differences in exposure levels in our sample were too small to result in statistically detectable differences in child outcomes. It is of course also possible that the true causal effect of NO2 exposure is smaller than the current literature suggests, or that negative effects only occur in settings where other protective factors (that may have been in place here) are not present. Further research will be needed to better understand the diverging results in the current literature.”

---

## [Decision Letter · Decision Letter 2]

25 Apr 2022

Assessing the association between air pollution and child development in São Paulo, Brazil

PONE-D-21-30740R2

Dear Dr. Fink,

We’re pleased to inform you that your manuscript has been judged scientifically suitable for publication and will be formally accepted for publication once it meets all outstanding technical requirements.

Kind regards,

Santosh Kumar

Academic Editor

PLOS ONE

**Santosh Kumar**

Associate Professor of Economics

Department of Economics and International Business

College of Business Administration

Sam Houston State University

1803 Ave I, Huntsville, Texas 77341-2056, USA

**P**: 001 (936) 294 2416; **F**: 001 (936) 294 3488

**Email**: skumar@shsu.edu

Academic Editor, PLOS Global Health

Research Fellow, Global Labor Organization (GLO)

Research Fellow, Institute for Labor Organization (IZA)

**Webpage**: https://sites.google.com/site/santoshkumar2987/

Reviewers' comments:

Reviewer's Responses to Questions

**Comments to the Author**

1. If the authors have adequately addressed your comments raised in a previous round of review and you feel that this manuscript is now acceptable for publication, you may indicate that here to bypass the “Comments to the Author” section, enter your conflict of interest statement in the “Confidential to Editor” section, and submit your "Accept" recommendation.

Reviewer #2: (No Response)

2. Is the manuscript technically sound, and do the data support the conclusions?

Reviewer #2: No

3. Has the statistical analysis been performed appropriately and rigorously? 

Reviewer #2: Yes

4. Have the authors made all data underlying the findings in their manuscript fully available?

Reviewer #2: Yes

5. Is the manuscript presented in an intelligible fashion and written in standard English?

Reviewer #2: No

6. Review Comments to the Author

Reviewer #2: (No Response)

7. PLOS authors have the option to publish the peer review history of their article (what does this mean?). If published, this will include your full peer review and any attached files.

Reviewer #2: No